# Co-Loaded Curcumin and Methotrexate Nanocapsules Enhance Cytotoxicity against Non-Small-Cell Lung Cancer Cells

**DOI:** 10.3390/molecules25081913

**Published:** 2020-04-21

**Authors:** Loanda Aparecida Cabral Rudnik, Paulo Vitor Farago, Jane Manfron Budel, Amanda Lyra, Fernanda Malaquias Barboza, Traudi Klein, Carla Cristine Kanunfre, Jessica Mendes Nadal, Matheus Coelho Bandéca, Vijayasankar Raman, Andressa Novatski, Alessandro Dourado Loguércio, Sandra Maria Warumby Zanin

**Affiliations:** 1Postgraduate Program in Pharmaceutical Sciences, Department of Pharmaceutical Sciences, State University of Ponta Grossa, 84030-900 Ponta Grossa, Brazil; loandacabral@hotmail.com (L.A.C.R.); pvfarago@gmail.com (P.V.F.); amandinhamlyra@gmail.com (A.L.); fer_barboza@hotmail.com (F.M.B.); traudiklein@gmail.com (T.K.); jessicabem@hotmail.com (J.M.N.); anovatski2@gmail.com (A.N.); aloguercio@hotmail.com (A.D.L.); 2Postgraduate Program in Pharmaceutical Sciences, Department of Pharmacy, Federal University of Paraná, 81020-430 Curitiba, Brazil; sandrazanin@ufpr.br; 3Postgraduate Program in Biomedical Science, Department of General Biology, State University of Ponta Grossa, 84030-900 Ponta Grossa, Brazil; cckanunfre@gmail.com; 4Postgraduate Program in Dentistry, Ceuma University, 65065-470 São Luís, Brazil; mbandeca@gmail.com; 5National Center for Natural Products Research, School of Pharmacy, University of Mississippi, University, MS 38677, USA; vraman@olemiss.edu

**Keywords:** Calu-3 cell line, cancer chemotherapy, drug resistance, poly(*ε*-caprolactone), poly(ethylene glycol)

## Abstract

*Background*: As part of the efforts to find natural alternatives for cancer treatment and to overcome the barriers of cellular resistance to chemotherapeutic agents, polymeric nanocapsules containing curcumin and/or methotrexate were prepared by an interfacial deposition of preformed polymer method. *Methods*: Physicochemical properties, drug release experiments and in vitro cytotoxicity of these nanocapsules were performed against the Calu-3 lung cancer cell line. *Results*: The colloidal suspensions of nanocapsules showed suitable size (287 to 325 nm), negative charge (−33 to −41 mV) and high encapsulation efficiency (82.4 to 99.4%). Spherical particles at nanoscale dimensions were observed by scanning electron microscopy. X-ray diffraction analysis indicated that nanocapsules exhibited a non-crystalline pattern with a remarkable decrease of crystalline peaks of the raw materials. Fourier-transform infrared spectra demonstrated no chemical bond between the drug(s) and polymers. Drug release experiments evidenced a controlled release pattern with no burst effect for nanocapsules containing curcumin and/or methotrexate. The nanoformulation containing curcumin and methotrexate (NCUR/MTX-2) statistically decreased the cell viability of Calu-3. The fluorescence and morphological analyses presented a predominance of early apoptosis and late apoptosis as the main death mechanisms for Calu-3. *Conclusions*: Curcumin and methotrexate co-loaded nanocapsules can be further used as a novel therapeutic strategy for treating non-small-cell lung cancer.

## 1. Introduction

Lung cancer is the deadliest of all cancers in the world. Sudden changes in lifestyle, environmental pollution, and smoking are strongly related to the development of lung cancer [1,2]. Treatment for lung cancer is typically guided by stage, although individual factors, such as overall health and coexisting medical conditions, are also important. Chemotherapy is beneficial at most stages of the disease, although it and radiation therapy are curative in only a minority of patients. The failure of chemotherapy in lung cancer treatment is mainly due to resistance mechanisms against chemotherapeutic agents, which result in a lack of therapeutic response [3,4]. Resistance phenomenon can occur by efflux pumps as P-glycoprotein (P-gp), which is expressed in human cells and acts as a localized drug transport mechanism, actively exporting drugs out of the cell [3,5,6]. In that sense, the efflux mechanism shows great clinical importance on lung cancer treatment and several compounds with P-gp inhibitor properties have been studied [7] to modulate chemotherapy resistance and to provide a more pronounced effect to chemotherapeutic drugs.

Curcumin (CUR) is a polyphenol obtained mainly from turmeric, the rhizome of *Curcuma longa* L. (Zingiberaceae), which is widely used in cooking as a spice and color additive due to its characteristic taste and deep yellow coloration [8,9]. CUR has demonstrated a variety of biological properties, such as antioxidant, anti-inflammatory, antimicrobial, and antitumor properties [8,10,11]. CUR has also shown therapeutic feasibility in improving wound healing activities by the use of nanotechnology-based delivery systems [11]. Moreover, CUR is a non-competitive inhibitor of P-gp by blocking the ATP hydrolysis process in efflux pump, which also paves the way for its use against lung cancer cells [7,12]. Despite these interesting effects, CUR has some disadvantages, such as its low aqueous solubility, low photostability, limited absorption, low bioavailability, rapid metabolism and elimination [13].

Methotrexate (MTX) is a well-known cytotoxic agent, which competitively and irreversibly inhibits dihydrofolate reductase, an enzyme that participates in tetrahydrofolate synthesis [14,15]. It is currently used in lung cancer treatment alone or along with other chemotherapeutic agents by oral and parenteral routes. High doses of MTX may circumvent at least two known mechanisms of resistance to this drug, membrane transport and high levels of the target enzyme. However, MTX can cause substantial toxicity by its systemic administration and can lead to remarkable side effects, such as hepatotoxicity, bone marrow depression, leucopenia, among others [16,17]. In that sense, the traditional use of this drug has two main problems when used against lung cancer: (a) its efflux from cancer cells by P-gp and (b) its high toxicity at usual therapeutic doses [18,19]. Thereby, new strategies to overcome these limitations are required.

Some controlled release strategies have been reported in the literature concerning the co-delivery of CUR and MTX. Dey et al. [20] developed gold nanoparticles containing CUR and MTX and evaluated their cytotoxic effect on C6 glioma cells and MCF-7 breast cancer cells. Curcio et al. (2018) [21] obtained pH-responsive polymersomes by self-assembling of a carboxyl-terminated PEG amphiphile achieved via esterification of PEG diacid with PEG40stearate. A highly hemocompatible co-delivery system of CUR and MTX was obtained. Vakilinezhad et al. [22] prepared PLGA nanoparticles for the co-administration of MTX and CUR as a potential breast cancer therapeutic system. Even though these authors reported a burst release from such nanoparticles, higher cytotoxicity was demonstrated against SK-Br-3 breast adenocarcinoma cell line. Curcio et al. [23] effectively delivered MTX to breast cancer cells by the use of a nanocarrier system derived from the self-assembly of a dextran-CUR conjugate prepared via enzyme chemistry with immobilized laccase acting as a solid biocatalyst. However, to the best of our knowledge, no previous paper was devoted to the preparation of co-loaded CUR and MTX nanocapsules using poly(ε-caprolactone) (PCL) as biodegradable polymer wall and poly(ethylene glycol) (PEG) as coating polymer focused on treating lung cancer.

Polymeric nanocapsules (NCs) are attractive colloidal systems to develop formulations containing labile and toxic substances. By definition, NCs are vesicular systems composed of a core, generally oily, surrounded by a polymer wall [24]. These carriers can present several advantages such as enhancing the dissolution process, increasing the therapeutic index, providing controlled delivery and achieving protection from the photo and chemical degradation [25]. Therefore, NCs can circumvent limitations provided by both CUR and MTX since they allow drug protection against degradation, improve bioavailability, and reduce possible side effects. In particular, NCs can reach target tissues, certain affected organs, and tumors due to their superior features as small particle size, large surface area, Brownian motion, and surface functionality which could provide higher cytotoxic effect even at low doses of the chemotherapeutic agents [11,13]. Moreover, the two-drug combination into NCs can produce a higher objective response since CUR can potentiate MTX activity by delaying its efflux from the lung cancer cells. 

Taking all these factors into consideration, this study was devoted to developing NCs for co-administration of CUR and MTX to provide a controlled release and a synergistic cytotoxic effect on non-small-cell lung cancer cell (Calu-3) growth. Moreover, in vitro studies were performed to evaluate the cytotoxic mechanism of these co-loaded NCs against Calu-3 cells.

## 2. Results and Discussion

### 2.1. Preparation and Characterization of Polymeric Nanocapsules (NCs) Containing Curcumin (CUR) and/or Methotrexate (MTX)

Nanocapsules with or without CUR and/or MTX were successfully obtained by the interfacial deposition of the preformed polymer method. Formulations containing no CUR showed a liquid aspect with a slightly bluish-white opalescent coloring. However, CUR-loaded nanocapsules presented a liquid aspect and an intense yellow color.

#### 2.1.1. Determination of Mean Diameter, Polydispersity Index, and Zeta Potential

Results of particle size, polydispersity, and zeta potential are summarized in Table 1. CUR and/or MTX-loaded and non-loaded NCs revealed mean sizes between 287.83 and 325.16 nm with a polydispersity index (PDI) varying from 0.290 to 0.351, which represent a certain system homogeneity. 

In general, NCs obtained by the interfacial deposition of preformed polymer method have demonstrated mean diameters between 200 and 300 nm and PDI between 0.2 and 0.3, particularly for the works carried out using poly(lactic-co-glycolic acid) and PCL [26]. However, NCs of larger diameter may be related to the presence of PEG in their composition. PEG chains create a more viscous organic phase, which affects its dispersion into the aqueous phase during stirring and leads to higher particle sizes with broader polydispersity [27].

Also, other aspects may directly influence the particle diameter in nanosystems. One of them is the oil used for preparing nanocapsules that influences some of the core structure properties, such as viscosity, hydrophobicity, and surface tension [28]. PDI values close to 0 are considered monodisperse and greater than 0.5 indicate heterogeneous dispersion [29]. Taking all these into account, suitable nanometric-scaled size and adequate polydispersity were recorded for NCs with or without CUR and/or MTX.

Negative zeta potential values were observed for all the colloidal suspensions of nanocapsules (Table 1). Zeta potential analysis allows identifying the electrical charges that occur on the surface of the nanoparticles. Particles are regarded as stable when their zeta potential values are higher than ± 30 mV [30]. Moreover, negative values were achieved for NCs on account of the anionic nature of PCL due to the presence of carboxylic acid functional groups [28]. The statistical analysis showed that mean diameter, PDI and zeta potential were similar for the formulations with or without CUR and/or MTX (*p* > 0.05).

#### 2.1.2. Encapsulation Efficiency

The encapsulation efficiency (EE) of CUR and/or MTX-loaded NCs are also depicted in Table 1. Mean EE values higher than 98.7% were obtained for formulations containing CUR alone or in combination with MTX. These values are attributed to the poor aqueous solubility of CUR (3.12 µg.mL^−1^ at 25 °C) [31], which avoided its partitioning in the aqueous phase. For those nanocapsules containing MTX, mean values varying from 82.4% (NCUR/MTX-2) to 88.9% (NMTX-1). These values are higher than 70%, hence suitable for nanocapsules containing lipophilic compounds as MTX (aqueous solubility of 2.60 mg·mL^−1^ at 20 °C) [32] when emulsion–diffusion methods were used [33]. These EE values were further used for obtaining final concentrations in µmol·L^−1^ during in vitro cell culture-based assays. For formulations containing both CUR and MTX, the EE compensation was performed using MTX due to its lower content in co-loaded NCs.

#### 2.1.3. Field Emission Scanning Electron Microscopy (FESEM)

Nanoscale dimensions were registered for CUR and/or MTX-loaded and non-loaded NCs when their images were assessed by FESEM (Figure 1). NCs were spherically-shaped and had a smooth surface. Their particle size and PDI were similar to those previously recorded [25] by photon correlation spectroscopy. Also, it was found that even after changing CUR and/or MTX concentrations, their morphologies were similar and no drug crystals were seen on their surfaces.

#### 2.1.4. X-ray Diffraction (XRD)

The diffractograms obtained for pure drugs (CUR and MTX), polymers (PCL and PEG 6000), and nanocapsules NCUR-1, NCUR-2, NMTX-1, NMTX-2, NCUR/MTX-0, NCUR/MTX-1, and NCUR/MTX-2 are shown in Figure 2. CUR presented typical peaks at 2θ values of 8.85°, 14.25°, 17.04°, and 23.10° as previously reported [34,35], which confirmed its crystalline nature. MTX showed main crystalline peaks at 9.16°, 12.8°, 19.42°, and 26.74° as previously described [36]. Also, PCL revealed two crystalline peaks at 21.59° and 23.72°. XRD data for PEG 6000 showed 2θ values at 19.17° and 23.41°.

CUR and/or MTX-loaded and non-loaded NCs exhibited similar non-crystalline pattern with a remarkable decrease of crystalline peaks from the drug(s) and polymers. In that sense, drug amorphization was demonstrated for all CUR and/or MTX-loaded formulations. This behavior is related to the molecular dispersion of such drugs when NCs were obtained using the interfacial deposition of preformed polymer method. Solid-state substances may have crystalline and/or amorphous characteristics. In general, the amorphous solids are more soluble than those in the crystalline state due to the free energies present in the dissolution process. In the amorphous state, molecules are randomly arranged and, therefore, lower energy is required to separate them, resulting in a faster dissolution when compared to the crystalline form [37]. Therefore, the amorphous pattern of CUR and/or MTX into NCs is considered to be advantageous because it allows the drug(s) dissolution and leads to drug(s) diffusion through the polymer wall, resulting in a sustained release of the encapsulated drug(s) [34,38].

#### 2.1.5. Fourier-Transform Infrared Spectroscopy (FTIR)

The FTIR spectra recorded for CUR, MTX, PCL, PEG 6000, physical mixture, and CUR and/or MTX-loaded and non-loaded NCs are depicted in Figure 3.

The FTIR spectrum of pure CUR presented the typical bands previously reported [35,39,40]. A sharp band at 3508 cm^−1^ and a broad band at 3387 cm^−1^ were assigned to the stretching of –OH group. Stretching vibrations of the carbonyl group and C–C bond of benzene ring were observed at 1627 cm^−1^ and 1510 cm^−1^, respectively. CUR showed a bending vibration at 1427 cm^−1^ assigned to –CH group connected to the benzene rings. A band at 1276 cm^−1^ was recorded for C–O stretching vibration. Two bands at 1155 cm^−1^ and 1028 cm^−1^ were assigned to stretching vibrations of C–O from ether group and C–O–C group, respectively. Bands at 960, 813, and 713 cm^−1^ were related to bending vibrations of the C–H bond of alkene groups. 

MTX presented bands at 3412 and 3387 cm^−1^ assigned to the stretching vibration of amine groups. A band at 2927 cm^−1^ was related to stretching of the O–H group from a carboxylic acid. The stretching of the C=O bond was observed at 1645 cm^−1^ while the stretching of N-H bong from amide was recorded at 1600 cm^−1^. Stretching bands at 1500, 1543, and 1448 cm^−1^ were assigned to the aromatic ring. A band at 1207 cm^−1^ was related to the stretching of the C–O bond from carboxylic acid [41].

PCL presented an intense band at 1730 cm^−1^, corresponding to the stretching of the C=O group from aliphatic esters. Two bands at 2943 and 2864 cm^−1^ were attributed to the asymmetric and symmetric stretching vibrations of –CH_2_ group, respectively. Stretching bands at 1174 and 1043 cm^−1^ were assigned to C–O bond.

The FTIR spectrum of PEG exhibited its typical broad band at 3520 cm^−1^, corresponding to the stretching of –OH group. In addition, two bands at 2883 cm^−1^ and 1110 cm^−1^ were assigned to –CH group from an aliphatic chain and asymmetric stretching of C–O–C from dialkyl ethers, respectively.

Bands at the same wavenumber range of FTIR spectrum to those recorded for the physical mixture was observed for CUR and/or MTX-loaded NCs. Considering the mild nanoencapsulation conditions used, no covalent functionalization occurred among polymers and drugs when formulations were obtained. 

### 2.2. In Vitro Drug Release Study

NCUR-2, NMTX-1, and NCUR/MTX-2 were initially chosen for in vitro drug release experiments due to their appropriate morphologies and sizes, suitable drug loadings and encapsulation efficiencies, and high amorphization of the drug(s) into NCs. The release profiles for free drugs and these NCs recorded by a dialysis method are summarized in Figure 4.

Free CUR and free MTX showed drug release patterns with a fast diffusion in the dissolution medium used. Free CUR achieved a mean drug-released value of 80% at 41 min of the experiment. For free MTX, a mean drug release of 80% was obtained at 195 min of the experiment. NCUR-2 released 80% of CUR at 2,640 min (44 h). NMTX-1 released 80% of MTX at 8,160 min (136 h). NCUR/MTX-2 had a mean drug release of 80% at 4,140 min (69 h) and 5,400 min (90 h), respectively for CUR and MTX. Therefore, NCs demonstrated prolonged release profiles with no burst effect in comparison to free drugs. Vakilinezhad et al. [22] described PLGA nanoparticles containing CUR and MTX with an initial burst release that was attributed to the drug dispersion into the superficial layers of these particles. In the present study, a typical core-shell structure was strategically planned to avoid this effect and to achieve a prolonged drug release for circumventing the administration of multiple doses per day of chemotherapy [42]. 

Considering the release profiles obtained for NCs, the difference in the release rate could be explained by the concentration gradient between each formulation and the dissolution medium. At higher drug concentrations, this process was faster since this gradient was the driving force for drug dissolution. Moreover, at higher drug-loading concentration, CUR and/or MTX could be distributed near to the nanocapsules surface providing a more rapid in vitro release [43]. In that sense, NCUR-2, prepared at the higher theoretical drug concentration of 3.0 mg·mL^−1^, presented a faster dissolution and reached a complete release in almost 2 days. On the other hand, NMTX-1, obtained at the lowest theoretical drug concentration of 0.1 mg·mL^−1^, revealed a very slow dissolution rate in about 7 days. The co-loaded formulation NCUR/MTX-2 presented the best dissolution features for controlling drug release. The theoretical drug concentrations proposed for NCUR/MTX-2, 0.5 mg·mL^−1^ of CUR and 0.3 mg·mL^−1^ of MTX, achieved a suitable intermediate dissolution rate in about 3 days. In this formulation, the release behavior of these drugs was closer as depicted in Figure 4, when compared to the other formulations tested. This performance is desired as it could ensure a simultaneous effect in vivo to provide both chemotherapeutic activity and inhibition of P-gp.

### 2.3. In Vitro Cell Culture-Based Assays

#### 2.3.1. Cell Viability by MTT and SRB Tests

NCUR-2, NMTX-1, and NCUR/MTX-2 were used for cell culture-based assays due to their known drug release profiles. In order to explore whether or not these formulations had a cytotoxic effect on the Calu-3 cell line, MTT and SRB assays were first performed. The cell viability results for NCUR-2 at high drug concentrations and NMTX-1 at low drug concentrations are shown in Table 2.

NCUR-2 significantly reduced Calu-3 viability at 200 and 100 µmol·L^−1^ for both MTT and SRB assays. This formulation also provided a significant reduction of cell viability at 50 µmol·L^−1^ for the SRB test. No statistically significant decrease was observed for other concentrations tested of NCUR-2 and for all concentrations investigated of NMTX-1.

In this study, CUR demonstrated cytotoxicity only at high concentrations. This effect may be related to the nanoencapsulation procedure since PCL/PEG NCs may have provided a controlled drug release during the 72 h assays, resulting in low concentrations of released CUR in contact within Calu-3 cells. Different results were observed when free CUR was used. Dhanasekaran et al. [44] investigated the effect of CUR on KG-1 cells using MTT reduction assay, where CUR achieved a dose-dependent reduction in cell viability at different exposure times. Cell viability was reduced to 57% at 50 µmol·L^−1^ for 24 h and to 20% at 100 µmol.L^−1^ for 48 h.

Although these data for pure CUR are more attractive, its free form is not a suitable option for its clinical use. CUR presents several problems related to its in vivo use due to its low aqueous solubility, low bioavailability, rapid hepatic metabolism, high decomposition rate at neutral or basic pH, and susceptibility to photochemical degradation. CUR generates inactive metabolites when administered orally, intraperitoneally, or intravenously, thus making it unfeasible to use in its free form [8].

In that sense, using CUR-loaded NCs is a viable and efficient therapeutic strategy in minimizing the negative features of CUR as well as to retain its antitumoral effect. Similar results have been reported for different tumor cell lines. Wang et al. [45] prepared solid lipid nanoparticles containing CUR and tested them on NCL-41299 and A549 cells. Yallapu et al. [46] obtained CUR-loaded polymeric nanoparticles and investigated their use against A2780CP and MDA-MB-231 lines.

Also, better results were obtained from the SRB assay instead of the MTT test. These findings were concerning to the high sensibility of SRB when cytotoxicity is investigated for cells growing adhered to dish bottom [47]. SRB evaluates cell viability through the ability of the dye to bind to protein components [47]. On the other hand, MTT examines the activity of mitochondrial dehydrogenase enzymes and their respective redox potential [48].

When cells were treated with NMTX-1, no significant reduction in Calu-3 viability was observed. This result is related to the very slow dissolution rate of MTX and the resistance phenomenon, in which MTX is a well-known substrate of P-gp [18,19]. Calu-3 is a tumor cell line that can express P-gp [49]. Therefore, lower intracellular accumulation of MTX was obtained since P-gp efflux pump was activated and led to the removal of the antitumor drug from within the cell.

To overcome this resistance mechanism, a co-loaded formulation (NCUR/MTX-2) was proposed since CUR even at low doses could block P-gp and could ensure the antitumor effect expected for MTX. The cell viability by MTT and SRB techniques of Calu-3, when treated with NCUR/MTX-2, is demonstrated in Figure 5.

NCUR/MTX-2 showed a significant reduction in Calu-3 cell viability by MTT procedure when 9.88:4 and 4.94:2 ratios of CUR:MTX were tested. For SRB assay, all concentrations (9.88:4; 4.94:2; and 2.47:1 ratio) provided a statistically significant decrease in tumor cell viability.

NMTX-1 at different drug concentrations of 1, 2, and 4 µmol·L^−1^ did not reduce Calu-3 viability when tested alone. However, co-loaded formulations containing the same MTX concentrations and low CUR concentrations achieved a statistical reduction of cell viability. This positive result may be due to the suitable dissolution profiles and the P-gp inhibition. NCUR/MTX-2 released both CUR and MTX in almost 72 h that was indeed the time interval during the in vitro cell culture-based assay was performed. Also, it is possible to infer that low CUR concentration in NCUR/MTX-2 plays a remarkable role as P-gp inhibitor, which allows that MTX remains within the cell and performs its antimetabolic/antitumor function. The effect of CUR was attributed only to its capacity of blocking P-gp because this drug demonstrated cytotoxicity against Calu-3 at concentrations higher than 50 µmol.L^−1^ as represented in Table 2. Dey et al. [20] studied alginate-stabilized gold nanoparticles containing CUR and MTX and investigated their effect on C6 glioma and MCF-7 cancer cells. Cell viability was reduced by the use of CUR:MTX ratios of 42:41 and 21:20.5 µmol·L^−1^. In this study, a statistically significant decrease in tumor cell viability was observed by the use of NCs containing CUR even at lower concentrations than those reported in the literature [13].

Another mechanism that may be responsible for the improved cytotoxicity of MTX by CUR is the increase of folate receptors (FRs) expression. FRs are transport systems that show a high affinity for folate and are expressed in large numbers of cancer cells. They are important for cell uptake of folic acid for its metabolism and cell division. However, MTX use can lead to cell resistance since this drug reduces FRs activity [50]. Furthermore, polysorbate 80, a pharmaceutical excipient used as a surfactant during NCs preparation, is a well-known P-gp inhibitor. Its molecules insert themselves between lipid tails of the lipid bilayer and fluidize the cell membrane. It may also interact with the bilayer’s polar heads and change the hydrogen bond or ionic bond forces which may contribute to its inhibitory action against P-gp [51]. In that sense, NC composition may enhance the effect of efflux pump inhibition provided by CUR and may lead to improved cytotoxicity against non-small-cell lung cancer cell growth even at low doses of CUR and MTX. 

Taking all these into account, the combination of low doses of CUR and MTX by the use of colloidal suspensions of NCs may be a promising strategy for the treatment of lung cancer, reducing the toxic effects associated with the use of high doses of MTX [44,50].

#### 2.3.2. Combination Index

The next question addressed was whether NCUR/MTX-2 could provide a synergistic effect on Calu-3 lung cancer cells using MTT cytotoxicity assay. To reveal the synergistic activity, a combination index (CI) was performed considering the half-maximal inhibitory concentration (IC_50_) data of separate and combined cytotoxic effects of NCs containing CUR and/or MTX. IC_50_ (mean value ± SD) for CUR from NCUR-2 was 100.18 ± 3.74 µmol.L^−1^. NMTX-1 presented an IC_50_ of 23.06 ± 1.13 µmol·L^−1^. IC_50_ for CUR and MTX from NCUR/MTX-2 were 69.81 ± 3.07 and 1.89 ± 0.06 µmol·L^−1^, respectively. Thereby, the CI value achieved was 0.78. In brief, the CI equal to 1 indicates that the two drugs have additive effects, the CI lower than 1 suggests a synergism and the CI higher than 1 indicates antagonism [52]. Consequently, a synergistic activity was demonstrated for NCUR/MTX-2 due to the combination of these drugs induced greater cytotoxicity against Calu-3. This finding shows that cytotoxic chemotherapy using NCUR/MTX-2 appears to be a promising strategy for the treatment of non-small-cell lung cancer which merits further preclinical and clinical investigation since it has allowed a suitable effect even at low chemotherapeutic doses into this nanocarrier.

#### 2.3.3. Cell Death Pattern through Acridine Orange/Ethidium Bromide (AO/EB) Test

Considering the results observed for NCUR/MTX-2 in reducing cell viability of Calu-3, the cell death pattern of this formulation was investigated after the 24 h treatment by AO/EB staining to verify the death mechanism.

The cytotoxic effect of NCUR/MTX-2 formulations can be observed in Figure 6. The control showed viable cells with normal and bright green nuclei (Figure 6A, white arrows). NCUR/MTX-2 resulted in early-stage apoptotic cells that were marked by crescent-shaped or granular yellow-green acridine orange nuclear staining (Figure 6B, yellow arrows), and late-stage apoptotic cells showing concentrated and asymmetrically located orange nuclear ethidium bromide staining (Figure 6B, blue arrow). In addition, the occurrence of apoptotic membrane blebbing (Figure 6B, red arrow) was verified since membrane blebbing is required for redistribution of fragmented DNA from the nuclear region into membrane blebs and apoptotic bodies [53]. No necrotic cell was observed after treating the Calu-3 cells with the co-loaded formulation.

Apoptosis is typically represented by a series of intracellular events, which ultimately lead to DNA fragmentation and the internucleosomal degradation of genomic DNA due to the activation of endogenous endonucleases. This mechanism is less aggressive because apoptosis is controlled and energy-dependent and can affect individual cells or clusters of cells [54]. On the other side, necrotic cells swell and rupture, releasing the cytoplasmic material. This process is characterized by injury to a group of cells rather than by individual death. Necrosis occurs by disruption of plasma membranes and organelles, marked dilation of mitochondria with the emergence of large amorphous densities, probably representing denatured proteins. The nucleus may shrink, suffer fragmentation or even totally disappear by unspecific DNA fragmentation [55]. In that sense, the co-loaded formulation was able to provide a Calu-3 death pattern by a carefully regulated energy-dependent process, characterized by programmed cell death, which is a widespread component of both health and disease. Moreover, apoptosis is a superior mechanism for feasible therapeutic interventions on the pathophysiology of lung cancer.

## 3. Materials and Methods 

### 3.1. Materials

Curcumin (CUR, ≥94% curcuminoid content, Sigma-Aldrich, St. Louis, MO, USA), methotrexate (MTX, >99% pure, Fermion, Espoo, Finland), poly(ε-caprolactone) (PCL, Mw 10,000–14,000 g.mol^−1^, Sigma-Aldrich), poly(ethylene glycol) 6000 (PEG, Mw 5,400–6,600 g.mol^−1^, Cromato Produtos Químicos, Diadema, Brazil), sorbitan monooleate (Span 80, Oxiteno, Mauá, Brazil), polysorbate 80 (Tween 80, Delaware, Porto Alegre, Brazil), medium chain triglycerides (MCT, 99% pure, Focus Química, São Paulo, Brazil), acridine orange base (AO, Sigma-Aldrich), ethidium bromide (EB, Sigma-Aldrich), methylthiazolyldiphenyltetrazolium bromide (MTT, Sigma-Aldrich), sulpho-rhodamine B (SRB, Sigma-Aldrich), penicillin-streptomycin (Sigma-Aldrich), and acetone (≥99.9% pure, Vetec Química, Rio de Janeiro, Brazil) were used as received. HPLC-grade methanol was purchased from Tedia (Rio de Janeiro, Brazil). RPMI 1640 medium and fetal bovine serum were obtained from Vitrocell (Campinas, Brazil). Water was purified in a Milli-Q Plus water purification system (Millipore, Bedford, MA, USA). All other solvents and reagents were analytical grade. The Calu-3 cell line was obtained from the Bank of Cells of Rio de Janeiro (BCRJ, Brazil) and was kindly provided by Dr. Katia Sabrina Paludo. 

### 3.2. Preparation of Polymeric Nanocapsules (NCs) Containing Curcumin (CUR) and/or Methotrexate (MTX)

The interfacial deposition of the preformed polymer method was used for preparing NCs containing CUR and/or MTX [20]. Six different formulations (Table 3) were obtained depending on the amount of CUR and/or MTX in their composition. Briefly, PCL and PEG 6000 were solvated in the organic phase containing span 80, CUR and/or MTX, MCT, and acetone. This phase was carefully added to the aqueous phase containing tween 80 and purified water under vigorous magnetic stirring at 45 °C. The obtained colloidal emulsion was kept under magnetic stirring for 10 min. The organic solvent was then quickly removed by evaporation under reduced pressure at 40 °C. Non-loaded nanocapsules were also prepared as the negative control. All formulations were obtained in triplicate from three different batches. For providing a comparative analysis, a physical mixture of polymers and drugs (PM PCL/PEG/CUR/MTX) at the same molar ratio was prepared by mortar and pestle mixing before characterization procedures.

### 3.3. Characterization of Polymeric Nanocapsules (NCs) Containing Curcumin (CUR) and/or Methotrexate (MTX)

#### 3.3.1. Determination of Mean Diameter, Polydispersity Index, and Zeta Potential of NCs

Mean particle size, polydispersity, and zeta potential were measured at 25 °C using a Zetasizer Nanoseries ZS903600 apparatus (Malvern Instruments, Malvern, UK). Each sample was previously diluted in water (1:500) and each analysis was performed at a scattering angle of 90° and a temperature of 25 °C for mean particle size and polydispersity measurements. For zeta potential, each sample was placed into the electrophoretic cell where a potential of ± 150 mV was used.

#### 3.3.2. Encapsulation Efficiency

CUR and/or MTX content of NCs was determined by the indirect method. In brief, suspensions of nanocapsules were submitted to a combined ultrafiltration/centrifugation using centrifugal devices (Amicon^®®^ 10.000 MW, Millipore, Bedford, MA, USA) at 2200 g during 30 min in triplicate. Free CUR and/or MTX were determined in ultrafiltrate using an HPLC method in a Merck-Hitachi Lachrom equipment (Tokyo, Japan), interface D-7000, UV detector module L-74000, equipped with pumps L-7100 and an integral degasser, controller software (Chromquest, Thermo Fisher Scientific, Incorporated, Pittsburgh, PA, USA), and manual injector (Rheodyne, Rohnert Park, CA, USA) equipped with a 20 µL injector loop and a 100 µL syringe (Microliter 710, Hamilton, Bonaduz, Switzerland). Chromatographic separation was accomplished using a Inertsil^®®^ ODS3 (GL Sciences, Torrance, CA, USA) reversed-phase analytical column (150 mm × 4.6 mm, 5 µm) and a GL Sciences Inertsil^®®^ ODS3 guard cartridge system (10 mm × 4 mm, 5 µm) at room temperature (20 ± 2 °C) using UV detection at 261 nm. Gradient elution was carried out using a mobile phase consisted of methanol:water acidified with 0.5% acetic acid at a flow rate of 1.0 mL.min^−1^. The mobile phase gradient program was as follows: it was started at 44% MeOH for 3 min, increased to 90% MeOH in 5 min, held constant until 11 min, then returned to 44% MeOH in 12 min. The encapsulation efficiency (EE, %) was calculated using Equation (1):(1)$$EE\left(\%\right)=\frac{total\;drug\;content-free\;drug\;content}{total\;drug\;content}\times 100$$

#### 3.3.3. Field Emission Scanning Electron Microscopy (FESEM)

The freeze-dried formulations were mounted on aluminum stubs and sputtered with gold (IC-50 Ion Coater, Shimadzu, Kyoto, Japan). Morphological analysis was performed and photomicrographs were prepared using a Mira3 LM FESEM (Tescan, Brno, Czech Republic) at an accelerating voltage of 5 kV.

#### 3.3.4. X-ray Diffraction (XRD)

Polymeric nanocapsules were previously deposited on a glass coverslip and dried at room temperature (25 ± 2 °C). XRD data were recorded in an Ultima IV diffractometer (Rigaku, Tokyo, Japan). The 2θ value was increased from 4° to 50° at a scan rate of 0.05°·min^−1^ using a Cu-Kα source (λ = 1.5418 Å) at 30 kV and 40 mA.

#### 3.3.5. Fourier-Transform Infrared Spectroscopy (FTIR)

FTIR analyses of raw materials, freeze-dried polymeric nanocapsules, and physical mixture were carried out from 4000 to 400 cm^−1^ using a Prestige-21 IR spectrophotometer (Shimadzu, Kyoto, Japan) in KBr pellets with 32 scans and a resolution of 4 cm^−1^.

### 3.4. In Vitro Drug Release Study

The release experiments were performed for free CUR, free MTX, and formulations NCUR-2, NMTX-1, and NCUR/MTX-2 by dialysis diffusion procedure in phosphate-buffered saline (PBS, 20 mmol.L^−1^) containing 0.1% (w/v) Tween^®®^ 80 at pH 7.4 [21,22] by adding each sample in PBS (1.5 mL) into a dialysis bag (Spectra/Por^®®^ molecular porous membrane tubing, MWCO 10,000, Spectrum Laboratories, Rancho Dominguez, CA, USA), and dialyzed against fresh PBS (13.5 mL) at 37 °C and under continuous magnetic stirring of 50 rpm. Aliquots of 500 µL were withdrawn at predetermined time intervals and replaced by the same volume of fresh medium. CUR and MTX concentrations were determined individually in each sample by the previously described HPLC method. The in vitro drug release assay was carried out in triplicate from three different batches.

### 3.5. In Vitro Cell Culture-Based Assays

#### 3.5.1. Cell Culture

Calu-3 cell line was cultured in RPMI 1640 medium at pH 7.4, containing 10% fetal bovine serum, supplemented with 24 mmol·L^−1^ sodium bicarbonate, 2 mmol·L^−1^ L-glutamine, 1 mmol·L^−1^ sodium pyruvate, 10,000 U·L^−1^ penicillin, and 10 mg·L^−1^ streptomycin. The cultures were maintained in a humidified oven at 37 °C with 5% CO_2_ atmosphere.

#### 3.5.2. Cell Treatment

Calu-3 cells were seeded in 96-well plates at a density of 1.5 × 10^4^ cells.well^−1^ and incubated for 24 h in the culture medium. For assessment of cell viability and cytotoxicity, Calu-3 cell line was incubated with the samples NCUR-2 (200.00–25.00 µmol·L^−1^), NMTX-1 (100.00–12.50 µmol·L^−1^ and 4.00–0.50 µmol·L^−1^) and NCUR/MTX-2 (247.00–61.75 µmol·L^−1^ and 9.88–2.47 µmol·L^−1^ to CUR; 100.00–25.00 µmol·L^−1^ and 4.00–1.00 µmol·L^−1^ to MTX) for 72 h at 37 °C with 5% CO_2_ atmosphere. For co-loaded formulation, final concentrations were standardized using MTX due to its lower content in NCs and EE compensation was also performed for CUR content, resulting in fractional values. Tests were obtained using serial dilution procedure and were performed as four independent experiments containing n = 4 samples per assay.

#### 3.5.3. Cell Viability by Methylthiazolyldiphenyl-tetrazolium Bromide (MTT) Test

After 72 h of the treatments, 200 μL of a solution of MTT at 0.5 mg·mL^–1^ was added to the wells following a standard method [56]. The cultures were then incubated at 37 °C for 2 h, protected from light, until the presence of formazan crystals. The supernatant was then removed. For the solubilization of these crystals, 200 μL of dimethyl sulfoxide was added. The spectrophotometric absorbance reading was performed at a wavelength of 550 nm in a μQuant microplate reader (BioTek, Winooski, VT, USA). To calculate cell viability (%), Equation (2) was used. The concentration that inhibited 50% of cell growth (IC_50_) was then calculated by Probit regression [57]:(2)$$Cell\;viability\;\left(\%\right)=\frac{absorbance\;of\;test}{absorbance\;of\;control}\times 100$$

#### 3.5.4. Combination Index

To define the drug-drug interaction potential from the use of NCs containing CUR and MTX against Calu-3 cells, the combination index (CI) was calculated by Equation (3) [58] considering the results obtained by the MTT method:(3)$$Combination\;index\;\left(CI\right)=\frac{\left(D\right)1}{\left(Dx\right)1}+\frac{\left(D\right)2}{\left(Dx\right)2}$$
where (Dx)1 and (Dx)2 are the concentration of the tested substance 1 (CUR) and the tested substance 2 (MTX) used in the single treatment that was required to decrease the cell viability by 50% and (D)1 and (D)2 are the concentration of the tested substance 1 (CUR) in combination with the concentration of the tested substance 2 (MTX) that together decreased the cell viability by 50%.

#### 3.5.5. Cell Viability by Sulphorhodamine B (SRB) Test

For the analysis of SRB, the method described by Papazisis et al. [59] was used. In brief, Calu-3 cell line was treated with each sample for 72 h and the supernatant was then discarded. The cells were washed with 200 μL of sodium phosphate buffer at pH = 7.4. Then, 200 μL of 10% cold trichloroacetic acid was added and the plates were placed in the refrigerator for 30 min to fix the cells. Subsequently, the cells were washed three times with 200 μL of distilled water and maintained for 24 h at room temperature (20 ± 2 °C) to dryness. Later, 200 μL of 0.2% SRB solution was added and kept for 30 min. The plates were then washed five times with 200 μL of 1% acetic acid and again dried for 30 min. Finally, 150 μL of 10 mmol.L^−1^ TrisBase was added and the spectrophotometric absorbance reading was performed at a wavelength of 432 nm in a microplate reader (μQuant, BioTek). Equation (2) was also used for calculating cell viability (%).

#### 3.5.6. Cell Death Pattern through Acridine Orange/Ethidium Bromide (AO/EB) Test

In 24-well plates, CALU-3 cells were seeded onto coverslips at the concentration of 1 × 10^5^ cells.well^−1^ using RPMI 1640 culture medium. After 24 h for cell adhesion, the medium was then discarded and NCUR/MTX-2 sample (CUR 9.88 µmol·L^−1^ and MTX 4.00 µmol·L^−1^) was added in a volume of 500 μL. Cells were then incubated at 37 °C in a wet atmosphere and 5% CO_2_.

After 24 h treatment, the wells were washed with 200 μL of sodium phosphate buffer at pH = 7.4. Following this, 10 μL of acridine orange/ethidium bromide dye mixture (200 μg·mL^−1^) was placed on a coverslip, which was then inverted on a slide. The staining was viewed under a BX41 fluorescence microscope (Olympus, Tokyo, Japan) with an excitation filter at 480/30 nm and emission at 535/40 nm. The typical fields were recorded with an attached Olympus DP71 camera.

Classification of cell type to AO/EB staining was performed based on the criteria proposed by Ribble et al. [60]. In summary, the viable cells present bright green nucleus, necrotic cells depict orange-red fluorescence, early-stage apoptotic cells are marked by crescent-shaped or granular yellow-green AO nuclear staining, and late-stage apoptotic cells show concentrated and asymmetrically sited orange nuclear EB staining.

### 3.6. Statistical Analysis

Statistical analyses were performed using one-way analysis of variance (ANOVA), and when necessary, the post-hoc Tukey test was used. The results were expressed as a mean ± standard error of the mean (SEM) and mean ± standard deviation (SD). P values lower than 0.05 (*p* <0.05) were considered significant. Calculations were carried out using the statistical software GraphPad Prism version 5.03 (GraphPad Inc., San Diego, CA, USA).

## 4. Conclusions

In conclusion, CUR and/or MTX co-loaded PCL/PEG6000 nanocapsules were successfully prepared by interfacial deposition of the pre-formed polymer. Nanometer-sized, monodisperse, amorphous/non-crystalline formulations with high drug-loading efficiencies were obtained. No changes in FTIR assignments were recorded after the nanoencapsulation procedure. In addition, NCUR/MTX-2 provided a statistically significant decrease of Calu-3 cell viability by MTT and SRB assays even at low concentrations of these chemotherapeutic agents. A synergistic effect was proven by the use of this co-loaded nanocarrier. The fluorescence-morphological analysis revealed a cell death pattern based on early and late apoptosis. No necrotic cell was observed in the Calu-3 cells after treating with the co-loaded formulation NCUR/MTX-2. In summary, the co-loaded nanocapsules method can be further used as a novel therapeutic strategy by intravenous or pulmonary route for treating non-small-cell lung cancer to hold the potential for achieving therapeutic outcomes while reducing the incidence of adverse drug effects.

## Figures and Tables

**Figure 1 molecules-25-01913-f001:**
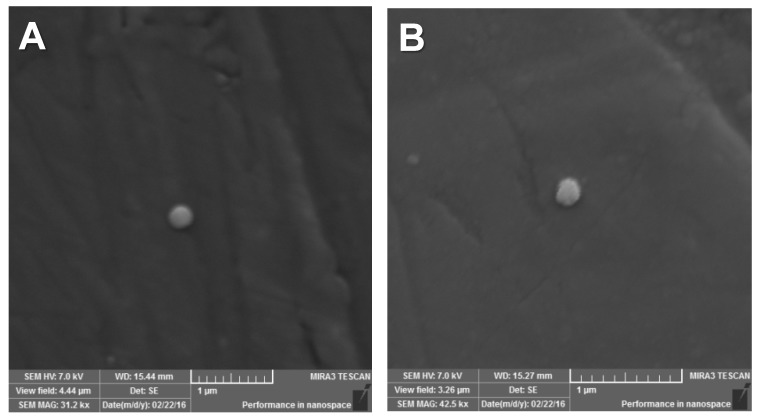
Photomicrographs of non-loaded and CUR and MTX-loaded NCs observed by FESEM: (**A**) NCUR/MTX-0 (31,200× magnification) and (**B**) NCUR/MTX-2 (42,500× magnification).

**Figure 2 molecules-25-01913-f002:**
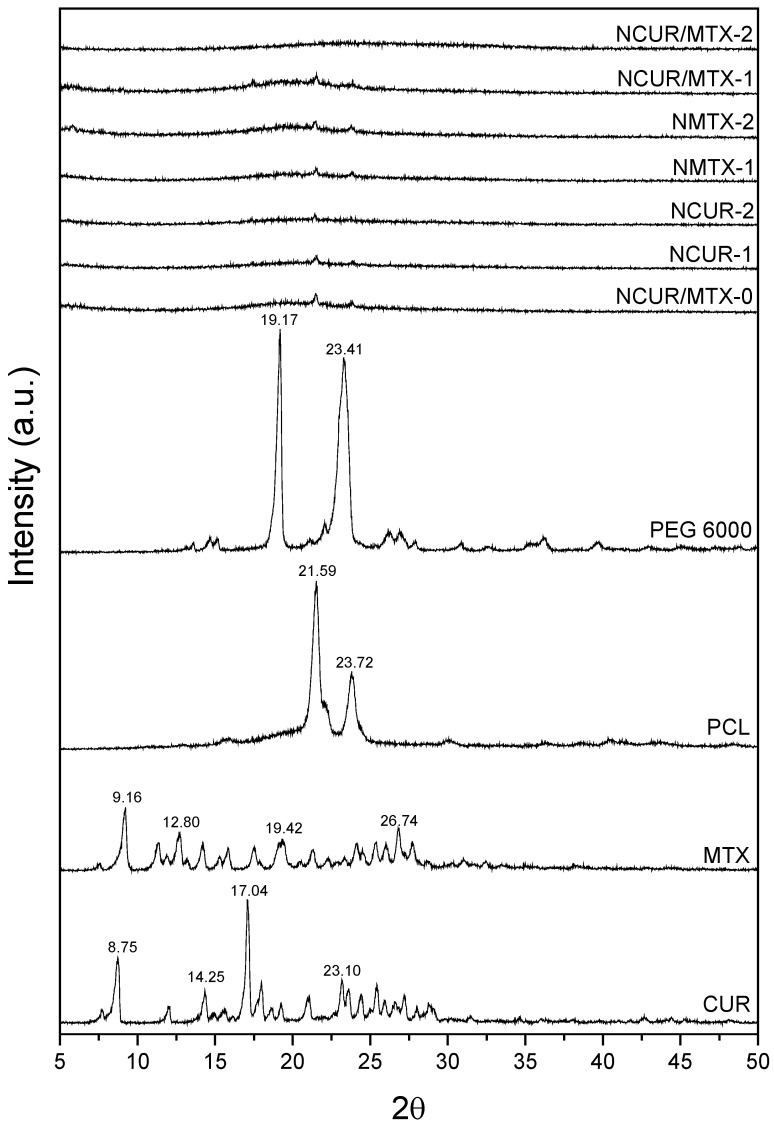
Diffractograms of CUR, MTX, PCL, PEG 6000, and CUR and/or MTX-loaded and non-loaded NCs obtained by XRD analysis.

**Figure 3 molecules-25-01913-f003:**
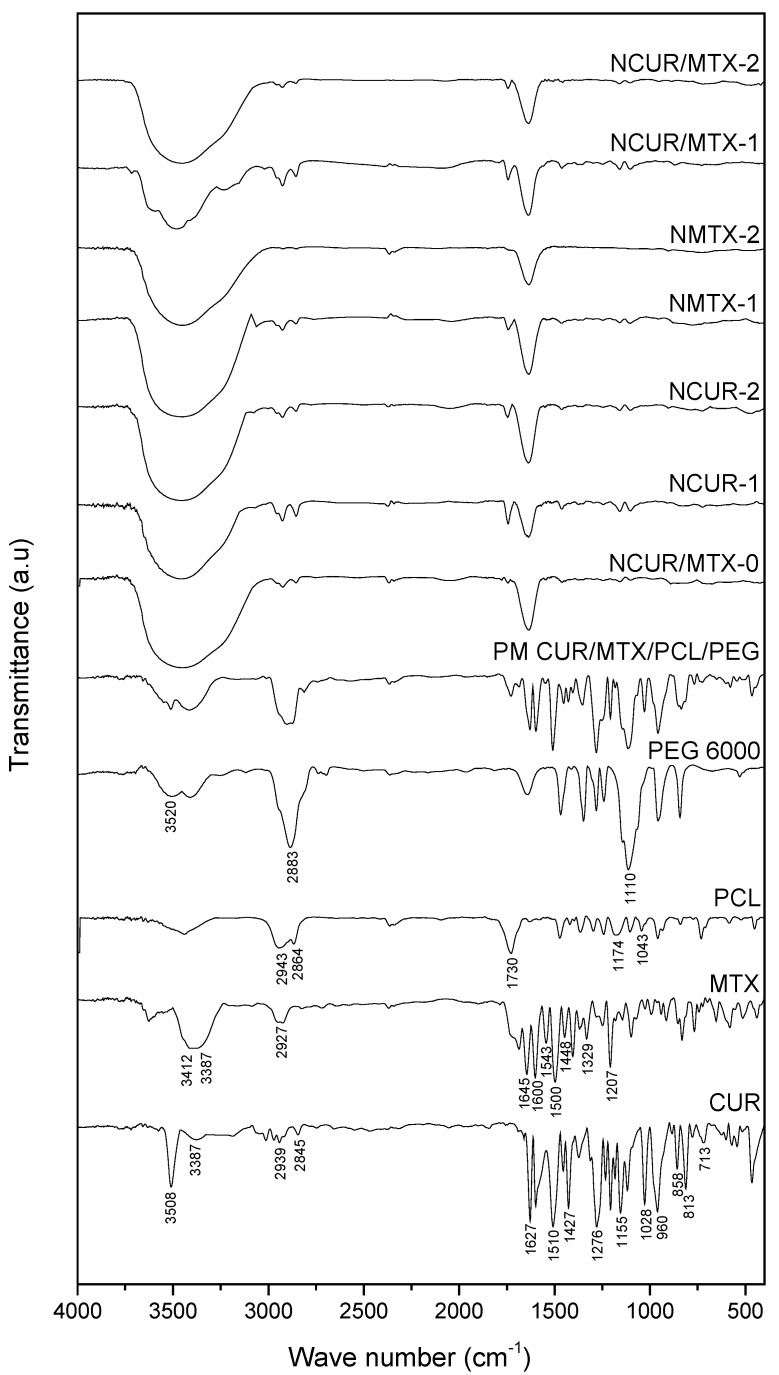
FTIR spectra of CUR, MTX, PCL, PEG 6000, physical mixture, and CUR and/or MTX-loaded and non-loaded NCs.

**Figure 4 molecules-25-01913-f004:**
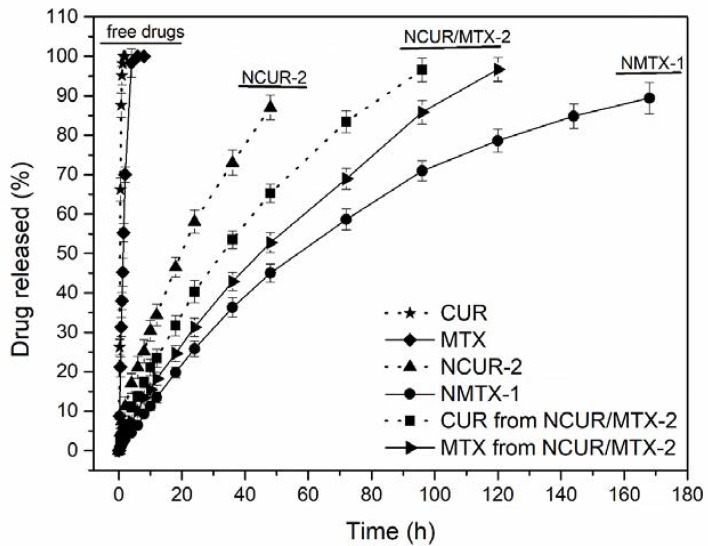
In vitro release profiles for free CUR, free MTX and nanocapsules (NCUR-2, NMTX-1, and NCUR/MTX-2).

**Figure 5 molecules-25-01913-f005:**
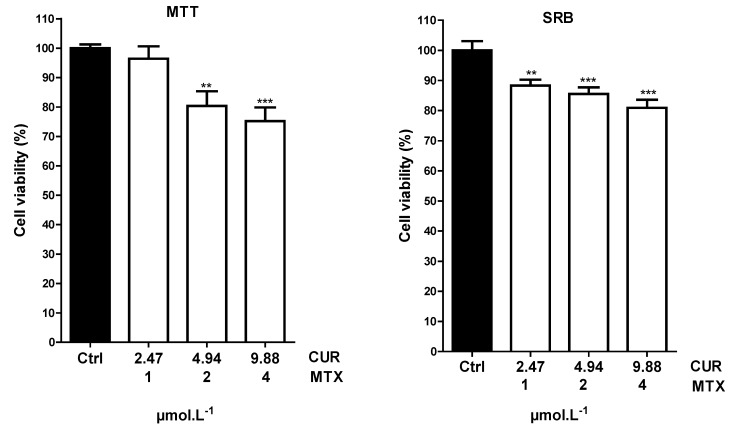
Cell viability of Calu-3 by MTT and SRB assays after its treatment with NCUR/MTX-2 at different (low) concentrations for 72 h. Results are expressed as mean ± standard error of the mean from 4 independent experiments. Asterisks denote significance levels compared to control: significantly different, ** *p* < 0.001 and *** *p* < 0.0001.

**Figure 6 molecules-25-01913-f006:**
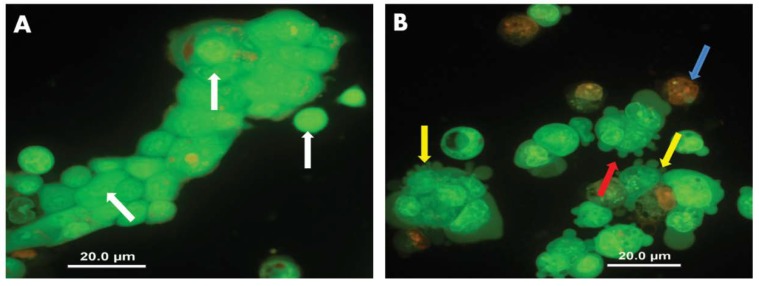
Effect of NCUR/MTX-2 on the morphology of Calu-3 cells stained with acridine orange/ethidium bromide by light fluorescence microscopy at 400× magnification. The negative control (**A**, vehicle) presented viable cells with normal nucleus staining represented by the bright green chromatin (white arrows); NCUR/MTX-2 (**B**) showed early-stage apoptotic cells that were marked by crescent-shaped or granular yellow-green acridine orange nuclear staining (yellow arrows), late-stage apoptotic cells showing concentrated and asymmetrically localized orange nuclear ethidium bromide staining (blue arrow), and apoptotic membrane blebbing (red arrow). Images are representative of results obtained from three independent biological replicates (n = 6 slides per sample).

**Table 1 molecules-25-01913-t001:** Mean diameter, polydispersity index (PDI), zeta potential, and encapsulation efficiency (EE) of the obtained CUR and/or MTX-loaded and non-loaded nanocapsules.

Formulation	Mean Particle Size * (nm)	PDI *	Zeta Potential * (mV)	EE * (%)
NCUR/MTX-0	313.30 ± 48.20	0.351 ± 0.15	−39.83 ± 2.46	-
NCUR-1	307.26 ± 29.40	0.323 ± 0.07	−38.80 ± 7.45	99.4 ± 0.51
NCUR-2	315.96 ± 10.30	0.351 ± 0.02	−41.20 ± 7.20	98.7 ± 0.66
NMTX-1	287.83 ± 28.00	0.290 ± 0.11	−40.60 ± 5.80	88.9 ± 0.47
NMTX-2	298.60 ± 25.00	0.292 ± 0.01	−38.96 ± 1.60	83.3 ± 0.42
NCUR/MTX-1	312.03 ± 23.60	0.344 ± 0.05	−39.43 ± 6.26	99.1 ± 0.63 (CUR) 84.4 ± 0.57 (MTX)
NCUR/MTX-2	325.16 ± 28.90	0.351 ± 0.03	−33.40 ± 3.29	99.3 ± 0.54 (CUR) 82.4 ± 0.44 (MTX)

* Values are depicted as mean ± standard deviation (SD).

**Table 2 molecules-25-01913-t002:** Cell viability of Calu-3 by MTT and SRB assays after its treatment with NCUR-2 and NMTX-1 at different concentrations for 72 h.

Sample	Final Concentration (µmol·L^−1^)	Cell Viability (%)	
		MTT	SRB
NCUR-2	**0**	100.00 ± 1.60	99.98 ± 1.75
	**25**	91.48 ± 3.36	88.30 ± 4.01
	**50**	87.74 ± 5.62	60.68 ± 6.10 ***
	**100**	49.91 ± 5.73 ***	33.87 ± 3.07 ***
	**200**	4.16 ± 0.67 ***	3.32 ± 0.47 ***
NMTX-1	**0**	100.00 ± 0.95	99.79 ± 1.22
	**0.5**	100.37 ± 1.96	96.10 ± 2.57
	**1**	100.60 ± 2.15	93.01 ± 4.48
	**2**	99.53 ± 1.67	90.85 ± 3.80
	**4**	93.90 ± 4.60	90.78 ± 4.67

Results are expressed as mean ± standard error of the mean from 4 independent experiments. Asterisks denote significance levels compared to control: significantly different, *** *p* < 0.0001.

**Table 3 molecules-25-01913-t003:** Composition of polymeric nanocapsules containing curcumin (CUR) and/or methotrexate (MTX).

Formulation	Composition								
	CUR (g)	MTX (g)	PEG (g)	PCL (g)	Span 80 (g)	MCT (g)	Acetone (mL)	Tween 80 (g)	Water (mL)
NCUR/MTX-0	-	-	0.020	0.080	0.077	0.300	27	0.077	53
NCUR-1	0.010	-	0.020	0.080	0.077	0.300	27	0.077	53
NCUR-2	0.030	-	0.020	0.080	0.077	0.300	27	0.077	53
NMTX-1	-	0.001	0.020	0.080	0.077	0.300	27	0.077	53
NMTX-2	-	0.003	0.020	0.080	0.077	0.300	27	0.077	53
NCUR/MTX-1	0.030	0.001	0.020	0.080	0.077	0.300	27	0.077	53
NCUR/MTX-2	0.005	0.003	0.020	0.080	0.077	0.300	27	0.077	53

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
