# Peer review of "Co-Loaded Curcumin and Methotrexate Nanocapsules Enhance Cytotoxicity against Non-Small-Cell Lung Cancer Cells"

_molecules, 2020, doi:10.3390/molecules25081913_

Round 1

Reviewer 1 Report

In this manuscript “Curcumin and methotrexate co-loaded nanocapsules enhance cytotoxicity against non-small-cell lung cancer cells”, the authors developed nanocapsules formulation with Curcumin and methotrexate for synergistic cytotoxic effect. This manuscript is suitable for publication in the journal “Molecules”, since the developed system has a higher cytotoxic activity and the properties of the formulation were evaluated with appropriate methods. Interestingly, for cytotoxic activity two methods MTT as well as SRB were used, that make a stronger case for this article. However, it can be published after substantial changes as mentioned below.

  1. Co-loaded Curcumin and methotrexate formulation is not a novel formulation. There are couple of articles that used this combination for higher cytotoxicity effect. So, ample emphasis should be provided to show this formulation superior in terms of cytotoxic activity than already published works.
  2. Nanocapsules or nanosuspension, one term should be maintained within the whole article.
  3. Page 3, Table 1, need to mention whether the results are presented with SEM or SD for better understanding.
  4. Page 4, Figure 1, in the description, FEG-SEM should be replaced with FESEM.
  5. Page 10, Line 289 AO/BE should be replaced with AO/EB.
  6. Need to mention the route of administration.
  7. Need to add the In-vitro drug release experiment.
  8. As the introduction part says it gives synergistic activity, to better analyze the data, and to highlight the dose-effect data of combined vs. single drug treatments Combination Indexes (CI) can be employed.

Author Response

April 7, 2020

Ms. Grace Zhang

Editor

Molecules

Dear Ms. Zhang,

            Sub.: Submission of the revised manuscript (ID: molecules-752742)

We are very grateful for the critical comments and helpful suggestions provided by the three anonymous reviewers. Their comments have greatly improved the final quality of our manuscript. We have made the necessary changes in the revised manuscript and highlighted the changes. Please find below our responses to all the reviewers’ comments.

Authors’ responses to the reviewer’s comments:

Reviewer 1

In this manuscript “Curcumin and methotrexate co-loaded nanocapsules enhance cytotoxicity against non-small-cell lung cancer cells”, the authors developed nanocapsules formulation with Curcumin and methotrexate for synergistic cytotoxic effect. This manuscript is suitable for publication in the journal “Molecules”, since the developed system has a higher cytotoxic activity and the properties of the formulation were evaluated with appropriate methods. Interestingly, for cytotoxic activity two methods MTT as well as SRB were used, that make a stronger case for this article. However, it can be published after substantial changes as mentioned below. 

  1. Co-loaded Curcumin and methotrexate formulation is not a novel formulation. There are couple of articles that used this combination for higher cytotoxicity effect. So, ample emphasis should be provided to show this formulation superior in terms of cytotoxic activity than already published works.

      Response: A new paragraph was added in the introduction to explain the differences between this article and other previous publications. In brief, the use of nanocapsules is innovative due to this system avoids burst effect observed for nanoparticles and shows the same properties as the Brownian motion, and the very small sizes provided higher cytotoxic effects even at low doses of curcumin and methotrexate. Besides, all the previously published works of this kind were focused on breast cancer. To the best of our knowledge, this is the first report focused on lung cancer treatment. This information was also added to the manuscript.

  1. Nanocapsules or nanosuspension, one term should be maintained within the whole article.

      Response: The term “nanocapsule” is now used throughout the article.

  1. Page 3, Table 1, need to mention whether the results are presented with SEM or SD for better understanding.

      Response: Results were presented as mean ± SD. This information was added to the manuscript.

  1. Page 4, Figure 1, in the description, FEG-SEM should be replaced with FESEM.

      Response: Done.

  1. Page 10, Line 289 AO/BE should be replaced with AO/EB.

      Response: Done.

  1. Need to mention the route of administration.

      Response: The cell culture-based assays were performed in vitro. However, the nanocapsules can be used by intravenous or pulmonary routes. This information was added to the manuscript.

  1. Need to add the In-vitro drug release experiment.

      Response: In vitro drug release experiment was performed as suggested by the reviewer. These data were then added to the manuscript in the methods, results, and discussion sections.

  1. As the introduction part says it gives synergistic activity, to better analyze the data, and to highlight the dose-effect data of combined vs. single drug treatments Combination Indexes (CI) can be employed.

      Response: Combination index (CI) was performed as requested. This information was added to the manuscript (methods, results, and discussion).

We have addressed all the comments made by the three reviewers and revised the manuscript accordingly. The manuscript was also revised by a native English speaker to correct the language and grammatical errors. All changes are highlighted in the manuscript. We have added Prof. Alessandro Dourado Loguercio and Mr. Matheus Coelho Bandéca as co-authors for their contributions in the newly included in vitro drug release study suggested by the reviewers.

We believe that the revised manuscript is acceptable for publication in the molecules. We remain at your disposal for any further clarification you may require regarding our manuscript. We look forward to hearing from you.

Thank you,

Best regards,

Dr. Jane Manfron Budel (Corresponding author)

Department of Pharmaceutical Sciences,

State University of Ponta Grossa,

4748, Carlos Cavalcanti Ave., 84030-900,

Ponta Grossa, Paraná, Brazil.

Email: janemanfron@hotmail.com

Reviewer 2 Report

This work reports a prep of polymeric nanocapsules containing curcumin to treat lung cancer cells. Evaluation of curcumin for cancer treatment possibility has been published many times. Regarding a nanocapsule, it is also not new in terms of synthesis, materials, or function. I don't find any originality nor advance in the field of cancer-targeting drug-delivery tools. Additional comments are listed below.

  1. Suggested nanocapsule does not have targeting moiety to lung cancer cells. How do the authors propose/validate their nanocapsule can travel toward lung in the body?
  2. The drug-release profile must be provided, at lease in in-vitro conditions.
  3. MTT assay depends on the absorption. Doesn't curcumin have an absorption in the range overlapping with the MTT reagent?
  4. Why are there no positive controls in the cell viability assays?
  5. Is there any meaningful difference between Figure 4 and Table 2?

Author Response

April 7, 2020

Ms. Grace Zhang

Editor

Molecules

Dear Ms. Zhang,

            Sub.: Submission of the revised manuscript (ID: molecules-752742)

We are very grateful for the critical comments and helpful suggestions provided by the three anonymous reviewers. Their comments have greatly improved the final quality of our manuscript. We have made the necessary changes in the revised manuscript and highlighted the changes. Please find below our responses to all the reviewers’ comments.

Authors’ responses to the reviewer’s comments:

Reviewer 2

  1. This work reports a prep of polymeric nanocapsules containing curcumin to treat lung cancer cells. Evaluation of curcumin for cancer treatment possibility has been published many times. Regarding a nanocapsule, it is also not new in terms of synthesis, materials, or function. I don't find any originality nor advance in the field of cancer-targeting drug-delivery tools.

      Response: A new paragraph was added to the introduction highlighting the novelties of the present study.

Additional comments:

  1. Suggested nanocapsule does not have targeting moiety to lung cancer cells. How do the authors propose/validate their nanocapsule can travel toward lung in the body?

      Response: The proposed formulation can be used by pulmonary and intravenous routes. This information was added to the manuscript. When the pulmonary route is used, the nanocapsules directly reach the pulmonary alveoli. When the intravenous route is used, these formulations were PEGylated to avoid immune recognition and to ensure a higher distribution all over the body. We performed an in vitro study as suggested by Reviewer 1, however, no in vivo validation was carried out. These experiments are currently in progress as part of another study.

  1. The drug-release profile must be provided, at lease in in-vitro conditions.

      Response: In vitro drug release experiment was performed as suggested by the reviewer. These data were then added to the manuscript (methods, results, and discussion).

  1. MTT assay depends on the absorption. Doesn't curcumin have an absorption in the range overlapping with the MTT reagent?

      Response: The absorption peak of curcumin is 434 nm and it has insignificant absorption above 500 nm. The MTT assay was performed at 550 nm. Therefore, curcumin provided no overlapping.

  1. Why are there no positive controls in the cell viability assays?

      Response: Because MTX is a well-known chemotherapy agent, no positive controls were used. Besides, several papers have reported the cytotoxic effect of
CUR. Therefore, this manuscript aimed to determine the lowest concentration that provides a suitable cytotoxic effect against Calu-3 cells.

  1. Is there any meaningful difference between Figure 4 and Table 2?

      Response: Yes. Table 2 depicts the data obtained for formulations containing only CUR or MTX. Figure 4 demonstrates the results observed for the CUR and MTX co-loaded formulation.

We have addressed all the comments made by the three reviewers and revised the manuscript accordingly. The manuscript was also revised by a native English speaker to correct the language and grammatical errors. All changes are highlighted in the manuscript. We have added Prof. Alessandro Dourado Loguercio and Mr. Matheus Coelho Bandéca as co-authors for their contributions in the newly included in vitro drug release study suggested by the reviewers.

We believe that the revised manuscript is acceptable for publication in the molecules. We remain at your disposal for any further clarification you may require regarding our manuscript. We look forward to hearing from you.

Thank you,

Best regards,

Dr. Jane Manfron Budel (Corresponding author)

Department of Pharmaceutical Sciences,

State University of Ponta Grossa,

4748, Carlos Cavalcanti Ave., 84030-900,

Ponta Grossa, Paraná, Brazil.

Email: janemanfron@hotmail.com

Reviewer 3 Report

Authors described preparation and drug loading of polymer nanoparticles based on PCL and PEG as well as their application in lung cancer treatment via in vitro experiment.

Comments:

Line 197 – The statement that “CUR and/or MTX-loaded NCs demonstrated bands at the same wavenumber range of FTIR spectrum to those recorded for the physical mixture. Therefore, these results suggest that there was no chemical bonding between drug(s) and polymers after nanoencapsulation” is rather speculative. FTIR analysis can hardly shows possibly forming new bonds (for example amides, esters,…) between drugs and already formed polymers used for encapsulation. I recommend to rewrite this sentence in meaning that at such mild encapsulation condition no covalent functionalization of drugs is expected…

Author Response

April 7, 2020

Ms. Grace Zhang

Editor

Molecules

Dear Ms. Zhang,

            Sub.: Submission of the revised manuscript (ID: molecules-752742)

We are very grateful for the critical comments and helpful suggestions provided by the three anonymous reviewers. Their comments have greatly improved the final quality of our manuscript. We have made the necessary changes in the revised manuscript and highlighted the changes. Please find below our responses to all the reviewers’ comments.

Authors’ responses to the reviewer’s comments:

Reviewer 3

Authors described preparation and drug loading of polymer nanoparticles based on PCL and PEG as well as their application in lung cancer treatment via in vitro experiment.

  1. Line 197 – The statement that “CUR and/or MTX-loaded NCs demonstrated bands at the same wavenumber range of FTIR spectrum to those recorded for the physical mixture. Therefore, these results suggest that there was no chemical bonding between drug(s) and polymers after nanoencapsulation” is rather speculative. FTIR analysis can hardly shows possibly forming new bonds (for example amides, esters,…) between drugs and already formed polymers used for encapsulation. I recommend to rewrite this sentence in meaning that at such mild encapsulation condition no covalent functionalization of drugs is expected…

      Response: The sentence was revised as recommended by the reviewer 3. The changes are highlighted in the manuscript.

We have addressed all the comments made by the three reviewers and revised the manuscript accordingly. The manuscript was also revised by a native English speaker to correct the language and grammatical errors. All changes are highlighted in the manuscript. We have added Prof. Alessandro Dourado Loguercio and Mr. Matheus Coelho Bandéca as co-authors for their contributions in the newly included in vitro drug release study suggested by the reviewers.

We believe that the revised manuscript is acceptable for publication in the molecules. We remain at your disposal for any further clarification you may require regarding our manuscript. We look forward to hearing from you.

Thank you,

Best regards,

Dr. Jane Manfron Budel (Corresponding author)

Department of Pharmaceutical Sciences,

State University of Ponta Grossa,

4748, Carlos Cavalcanti Ave., 84030-900,

Ponta Grossa, Paraná, Brazil.

Email: janemanfron@hotmail.com

Round 2

Reviewer 1 Report

The corrections have been made as my recommendation, which makes it a better article. The manuscript is suitable for publication in the journal “Molecules” as is.

Reviewer 2 Report

The authors answered my previous comments.